# Carbon Dioxide Pretreatment and Cold Storage Synergistically Delay Tomato Ripening through Transcriptional Change in Ethylene-Related Genes and Respiration-Related Metabolism

**DOI:** 10.3390/foods10040744

**Published:** 2021-04-01

**Authors:** Me-Hea Park, Sun-Ju Kim, Jung-Soo Lee, Yoon-Pyo Hong, Seung-Hun Chae, Kang-Mo Ku

**Affiliations:** 1Postharvest Research Division, National Institute of Horticultural & Herbal Science, Wanju 55365, Korea; lij808@korea.kr (J.-S.L.); hongyp0113@korea.kr (Y.-P.H.); 2Department of Bio-Environmental Chemistry, Chungnam National University, Daejeon, Chungnam 34134, Korea; kimsunju@cnu.ac.kr; 3Department of Horticulture, College of Agriculture and Life Sciences, Chonnam National University, Gwangju 61186, Korea; kjcmc0921@naver.com; 4Interdisciplinary Program in IT-Bio Convergence System, Chonnam National University, Gwangju 61186, Korea

**Keywords:** carbon dioxide, chilling injury, ethylene response factor, ripening, tomato

## Abstract

The effects of CO_2_ pretreatment before cold storage on tomato quality were investigated using physicochemical and transcriptome changes. Harvested tomatoes were treated with 30% or 60% CO_2_ for 3 h before storage at 4 °C for 14 d (cold storage), followed by transfer to 20 °C for 8 d (ambient conditions). The CO_2_-treated fruits were firmer with a better appearance than untreated fruits, even after being transferred from 4 °C storage to 20 °C for 8 d. CO_2_ pretreatment coupled with cold storage synergistically delayed tomato ripening by reducing respiration and lowering lycopene production. The tomatoes treated with 30% and 60% CO_2_ had fewer pits than untreated fruits after cold storage, even after being transferred to ambient conditions. Moreover, the 60% CO_2_ treatment significantly suppressed the decay rate. Transcriptome and metabolome functional enrichment analyses commonly showed the involvement of CO_2_-responsive genes or metabolites in sucrose and starch metabolism, as well as biosynthesis of secondary metabolites—in particular, glycolysis reduction. The most frequently detected domain was the ethylene-responsive factor. These results indicate that altered ethylene biosynthesis and ethylene signaling, via ethylene-responsive transcription factors and respiration-related pathways, appear to control CO_2_-induced fruit quality.

## 1. Introduction

Tomato (*Solanum lycopersicum* Mill.) is an important crop, both nutritionally and commercially, as it serves as a good source of fiber, vitamins, beta-carotene, and lycopene. However, tomatoes have a relatively short postharvest shelf life owing to their rapid ripening rate and sensitivity to cold-storage conditions, which limit their transportability and marketability. During storage and transportation, ripening progresses with a color change from green to red, as well as softening and compositional changes in chemicals related to flavor and aroma, such as organic acids, sugars, and volatile compounds. Cold storage can maintain vegetable freshness for long periods by suppressing respiration, but it also induces a physiological disorder known as chilling injury (CI), which occurs when fruits are held at a critical temperature for too long [1]. CI in tomatoes is typified by pitting, the development of sunken areas on the fruit (blemishes), and increased susceptibility to rot and decay induced by *Alternaria* spp. [2]. These CI symptoms appear when the fruits are subjected to ripening temperatures (20–22 °C) after cold storage (2–6 °C) for more than 2 weeks [3]. Hence, CI symptoms usually become pronounced under market-shelf conditions following cold storage, thus reducing consumer desirability [4]. However, when exporting tomatoes overseas, a long-term cold-chain transportation system is required; therefore, CI is a serious hurdle. Other causes of postharvest losses in tomatoes include softening, cracking, black mold rot, and gray mold rot. Thus, practical and feasible techniques to extend the shelf life of tomatoes by reducing postharvest losses and CI are required.

The benefits of exposure to high levels of carbon dioxide (CO_2_) as pre- and postharvest treatments have been investigated in several commodities [5,6,7]. Treatment comprising exposure to high levels of CO_2_ effectively controlled postharvest diseases and enhanced firmness in strawberries. This resulted in an extended shelf life without altering the color, solid soluble content (SSC), titratable acidity (TA), or pH of the fruit [8,9,10]. Moreover, it decreased astringency in persimmon [5,11], inhibited browning, and improved the quality of fresh-cut burdock owing to a reduction in respiration rate and polyphenol oxidase [7]. Cell wall-degrading enzyme activity was altered upon exposure to high concentrations of CO_2_ in postharvest ’Mihong’ peach, and the delayed softening resulted in a reduction in its decay rate after 4 d of storage at 23 °C [12]. Furthermore, recent studies on postharvest management of tomatoes showed that high levels of CO_2_ reduce the decay rate of cherry tomatoes [13].

CO_2_ treatment improves tolerance to prolonged cold storage. Ezz et al. [14] suggested that treatment with high levels of CO_2_ reduced chilling-induced peel pitting in grapefruit by controlling proline metabolism. Additionally, prestorage CO_2_ treatment at 10–40% was found to reduce CI symptoms in citrus fruit [15], and CO_2_ treatment before storage at 2 °C effectively reduced chilling-induced physiological changes in zucchini [16]. Bang et al. [10] showed the cellular responses of strawberry fruit treated with CO_2_ using an integrated transcriptomic-metabolomic analysis. Transcription analysis of short-term, CO_2_-treated table grapes before cold storage showed that CO_2_ treatment seems to be an active process requiring the activation of transcription factors, as well as protein kinases, in early harvest table grapes [17]. Transcriptome analysis of ‘Wonderful’ pomegranate fruit showed changes in transcripts related to metabolic pathways involving primary and secondary carbohydrate metabolism [18]. Recently, integrative analysis of metabolite and transcript profiles revealed a complex regulatory network in tomatoes under chilling stress [19,20].

Nevertheless, at the molecular level, little is known regarding the response of tomato to CO_2_ treatment during postharvest cold storage. Integrated transcriptomic-metabolomic analysis provides a better understanding of CO_2_ effect on quality, including CI, in tomato.

Therefore, in this study, we aimed to develop a practical postharvest technique to extend the shelf life of tomatoes. To do this, we investigated the effects of short-term exposure to CO_2_ on tomato fruit quality and found that CO_2_ treatment delayed ripening and reduced CI symptoms. Furthermore, to improve our understanding of the molecular mechanisms involved in the beneficial effects of CO_2_ treatment on tomato quality, comparative transcriptomic and metabolite analyses between CO_2_-treated and untreated fruits were performed. The results obtained improve our understanding of the manner in which tomato responds to CO_2_ treatment and how tomato fruit quality can be maintained during postharvest storage.

## 2. Materials and Methods

### 2.1. Plant Materials and Treatments

’Defunis’ tomato fruits between the mature-green and breaker stages were harvested in summer at Jungyeum, South Korea. Immediately after their transport to the laboratory, the fruits were treated with 30% or 60% CO_2_ (mixed with ambient air) for 3 h in a closed chamber. For the control and treatment groups, tomatoes were left in ambient air using 30 commercial cardboard boxes in each group (each box contained 30 fruits). The CO_2_ concentration in the closed chamber was measured using a portable headspace analyzer (Dansensor, Ringsted, Denmark). The control samples were flushed with ambient air only. The chamber was subsequently flushed with air to remove the CO_2_. The damaged fruits were discarded. After the treatment, both control and CO_2_-treated tomato boxes were covered with 0.03 mm low-density polyethylene film and stored at 4 °C (cold storage) for 14 d or 4 °C for 14 d followed by 8 d at 20 °C (14 + 8; storage at ambient conditions at 20 °C). The relative humidity was maintained at 90 ± 5% during storage.

### 2.2. Gas Chromatography Analysis

Respiration and ethylene production were analyzed using gas chromatography (Bruker 450-GC; Bruker Corp., Billerica, MA, USA) as described by Park et al. [13]. One milliliter of gas was sampled using a syringe from a 2 L container with four fruits from each treatment, which were previously sealed for 2 h. The injection and column temperatures were 110 °C and 70 °C, respectively. The thermal conductivity detector and flame ionization detector used for the CO_2_ and ethylene measurements were set at 150 °C and 250 °C, respectively.

### 2.3. Fruit Quality Evaluation

Fifteen fruits per treatment were sampled to assess the fruit quality. Skin color was monitored using a color meter (Minolta CR-400; Konica Minolta, Osaka, Japan), and values were reported based on the Hunter’s scale, redness (a*). Firmness was analyzed using a texture analyzer (TA Plus Lloyd Instruments Ltd., Fareham, Hampshire, UK) at a speed of 2 mm/s with a plunger head of 5 mm in diameter. The total SSC of the samples was analyzed using a digital refractometer (PAL-1; Atago Co., Ltd., Tokyo, Japan), and TA was determined by titrating 5 mL of juice from one fruit with 0.1 N NaOH until a pH of 8.2 was reached. This procedure was performed using an auto-pH titrator (Titroline Easy; SCHOTT Instruments GmbH, Mainz, Germany), and the TA was expressed in grams of citric acid per 100 g of sample juice. CI was measured as described by Park et al. [21]: 0 = no pitting, 1 = few, scattered pits, 2 = pitting covering up to 5% of the fruit surface, 3 = pitting covering 5–25% of the fruit surface, and 4 = extensive pitting covering >25% of the fruit surface. Fruit decay was expressed as the percentage of fruits showing decay symptoms. The CI index and decay rate were recorded as three replicates (three boxes of 30 fruits each) per treatment, per day.

### 2.4. Carotenoid Analysis

Carotenoid analysis was performed as described by Sadler et al. [22]. Carotenoids were extracted from 500 mg of dried powder samples, obtained from pericarp tissues, using hexane, acetone, and ethanol (2:1:1). The hexane layer was collected, and the concentrated solution was adjusted to 2 mL (*v*/*v*) with methyl tert-butyl ether and filtered for analysis. Carotenoids were quantified using an HPLC Agilent 1200 series system (Agilent Technologies Inc., Santa Clara, CA, USA) equipped with a Kinetex C18 100A column (100 × 4.60 mm, 2.6 μm; Phenomenex Inc., Torrance, CA, USA). The HPLC conditions were as follows: column temperature, 40 °C; detection wavelength, 454 nm; flow rate, 0.8 mL/min; and injection volume, 20 μL. Carotenoids were analyzed via gradient elution (70–100%) of mobile phase solvents A (water:methanol = 25:75 (v/v)) and B (ethyl acetate). Compounds were identified by comparing their elution times with those of the verified standards.

### 2.5. Transcriptome Analysis

Tomato fruits were sampled from the untreated control, as well as from the 30% and 60% CO_2_-treated groups, after 0 d, 7 d, and 14 d of storage at 4 °C, and 8 d of storage at 20 °C. Subsequently, five fruits were pooled from each sample, and the pericarp tissue was used for RNA isolation using the cetyltrimethylammonium bromide protocol [23]. Library preparation and RNA sequencing (RNA-Seq) were performed by Macrogen (Seoul, Korea). Processed reads were aligned to Solanum lycopersicum (GCF_000188115.3_SL2.50) using HISAT v.2.0.5(1) [24]. After alignment, StringTie v.1.3.3b [24] was used to assemble the aligned reads into transcripts and estimate their abundance. The expression level of each transcript was normalized to the values of fragments per kilobase of exon per million fragments mapped (FPKM). The filtered data were log2-transformed and subjected to quantile normalization. Differentially expressed genes (DEGs) were selected using *p* ≤ 0.05, whereas log_2_-fold change values (FC) ≥ 2 were used as thresholds. For the DEG set, gene enrichment, functional annotation, and pathway analyses were performed using the DAVID tool (http://david.abcc.ncifcrf.gov/, accessed on 22 March 2018) and the Kyoto Encyclopedia of Genes and Genomes (KEGG) database (http://www.genome.jp/kegg/pathway.html, accessed on 22 March 2018). To provide a functional overview of DEGs between the CO_2_ treatment and non-treatment groups, DAVID analysis was performed using all the CO_2_-responsive genes, and conserved domains in DEGs were analyzed with InterPro using the DAVID tool. Hierarchical clustering analysis was performed using complete linkage and Euclidean distance as a measure of similarity to display the expression patterns of differentially expressed transcripts with FC ≥ 2. Data analyses and visualization of DEGs were performed using R v.3.4.3 (www.r-project.org, accessed on 22 March 2018). Expression profiling of the DEGs involved in ethylene signaling and synthesis was performed using the PermutMatrix software [25].

### 2.6. Quantitative Real-Time PCR (qRT-PCR)

Quantitative real-time PCR (qRT-PCR) was performed as described by Park et al. [21] using a CFX96 TouchTM Real-Time PCR detection system (Bio-Rad, Hercules, CA USA). Amplification was performed using iQTM SYBR Green Supermix (Bio-Rad) with specific primers (Appendix A). qRT-PCR was performed under the following conditions: 95 °C for 30 s, followed by 40 cycles of 95 °C for 10 s and 55 °C or 58 °C for 40 s. Relative gene expression was calculated using the ^ΔΔ^Ct method and normalized using the expression levels of the housekeeping genes, *actin* and *elongation factor 1* (*EF1*). qRT-PCR analysis was performed using at least three biological replicates and two technical replicates.

### 2.7. Water-Soluble Primary Metabolite Profiling Using Gas Chromatography-Mass Spectrometry (GC-MS)

Water-soluble primary metabolites were tested using the methods described by Lisec et al. [26]. Fifty milligrams of tomato powder were weighed into 2 mL tubes, which were then vortexed for 10 min with 1.4 mL of methanol from the freezer, using 50 µL of 10 mg/mL ribitol as an internal standard. Subsequently, the tubes were centrifuged at 10,000× *g* for 3 min. Supernatant (700 µL) from the centrifuged sample was transferred to 2 mL microcentrifuge tubes. The supernatant was vortexed for 10 s with 700 µL of H_2_O. Subsequently, 10 µL of the extract solution was transferred to 1.5 mL tubes. The samples were placed in a speed vac (Vision, Bucheon, Gyeonggi-do, Korea) at 30 °C for 1 d. After drying, the samples were centrifuged at 800× *g* for 90 min at 37 °C with 50 μL of a freshly prepared mixture of 40 mg/mL methoxyamide in pyridine. The sample was then centrifuged at 800× *g* for 20 min at 50 °C with 80 μL of N-methyl-N-(trimethylsilyl)trifluoroacetamide. For water-soluble metabolite analysis using GC-MS, the GC oven was set at an initial temperature of 80 °C for 2 min. Subsequently, the oven temperature was increased by 15 °C per min up to 330 °C and maintained for 5 min. The injector and detector temperatures were set at 205 °C and 250 °C, respectively. An aliquot (1 μL) of the sample was injected at a split ratio of 200:1, and the carrier gas (helium) was maintained at a constant flow rate of 1.2 mL/min. The mass spectrometer was operated in the positive electron impact mode at an ionization energy of 70.0 eV and a scan range of 40–500 *m*/*z* [27].

### 2.8. Statistical Analyses

Values are presented as the mean ± standard error. Samples were subjected to analysis of variance (ANOVA), and significant differences were determined using Duncan’s multiple range test or Tukey’s Honestly Significant Difference (HSD) test. Partial least squares-discriminant analysis (PLS-DA) and pathway analysis were performed using MetaboAnalyst (https://www.metaboanalyst.ca/, accessed on 19 March 2021). All analyses were performed using SAS v.9.2 (SAS Institute, Cary, NC, USA).

## 3. Results

### 3.1. Respiration and Ethylene Production

Respiration rates were higher in tomatoes treated with 30% and 60% CO_2_ than in the control fruits, indicating the successful absorption of CO_2_ in the pretreated tomatoes (Figure 1A). However, with the progression of time, respiration rates decreased, with some fluctuations, and reached the control level by the end of the cold-storage period. Upon transferring tomatoes to 20 °C, respiration rates increased and reached the highest value at 4 d, declining thereafter.

Before cold storage, ethylene production was higher in the CO_2_-treated fruits than in the untreated fruits (Figure 1B, inner box). However, this difference decreased on the second day of cold storage, and no significant difference was found among the control, 30% CO_2_-treated, and 60% CO_2_-treated groups of fruits during cold storage. However, 4 d after transfer to the storage at ambient conditions, CO_2_-treated fruits exhibited lower ethylene production than the control fruits (Figure 1B).

### 3.2. Effect of CO_2_ on Fruit Quality and Ripening of Tomatoes

Analysis of SSC or TA showed that CO_2_ treatment had no significant effect on fruit eating quality. Nevertheless, the pH of the treated fruits was higher than that of the untreated fruits after cold storage and after transfer to 20 °C for 4–8 d (Table 1). Fruit firmness decreased with storage time; however, tomatoes treated with 30% or 60% CO_2_ were significantly firmer than the control tomatoes, even after 8 d of storage at ambient conditions (Table 1). Notably, the difference in firmness was observed only in storage at ambient conditions but not during cold storage. Tomato is a climacteric fruit, which follows a ripening pattern that is controlled by ethylene; therefore, ripening is delayed if ethylene production is inhibited [13]. As shown in Figure 1B, ethylene production in CO_2_-treated tomatoes was lower than that in the control during storage at ambient conditions, suggesting that lower ethylene production in CO_2_-treated tomatoes led to a delay in tomato softening. There were no significant differences between the 30% and 60% CO_2_-treated tomatoes in terms of firmness, SSC, and TA.

Prestorage short-term exposure to CO_2_ treatment delayed the ripening of tomatoes stored at 4 °C for 14 d, even after they were transferred to 20 °C for 8 d. Color is one of the most important visual attributes in the ripening index of tomatoes [21]. Fruits treated with 30% and 60% CO_2_ had a lower a* than the untreated fruits during 14 d of cold storage but not after transferring to 20 °C (Figure 2A,B). The color change during the ripening of the tomato fruit is due to the degradation of chlorophyll, coupled with the synthesis of different anthocyanins and the accumulation of carotenoids such as β-carotene, xanthophyll esters, xanthophylls, and lycopene. In particular, lycopene accumulation is correlated with tomato fruits [28]. To verify this and determine the influence of CO_2_ on ripening, we investigated the carotenoid content of tomatoes during the storage period. As expected, the treated tomatoes showed lower lycopene content than the untreated tomatoes. In contrast, CO_2_ treatment enhanced the beta-carotene content of tomatoes at the beginning of storage, whereas lutein content decreased with the progression of storage (Table 2). Under cold storage, the CO_2_-treated tomatoes showed significantly lower lycopene content than the control. However, beta-carotene and lutein content was higher in the treated tomatoes than in the control. These results are similar to those reported by Park et al. [21], who showed that cold storage inhibited lycopene synthesis in tomatoes. Under storage at ambient conditions, lycopene content increased dramatically in coordination with color change in all tomatoes; however, lycopene content was lower in the CO_2_-treated fruits than in the controls (Figure 2B). This suggests that CO_2_ treatment delayed ripening and extended the shelf life of tomatoes, and that these effects were synergistic under cold-storage conditions.

### 3.3. Effect of CO_2_ on Chilling Injury and Decay of Tomatoes

CI symptoms were observed in the control fruits after 7 d at 4 °C and increased with the progression of cold storage for 14 d, followed by a further 8 d at 20 °C (Figure 2C). A previous report showed that mature, green tomatoes stored below 12.5 °C for longer than 2 weeks showed CI symptoms such as surface pitting and increased fungal growth during subsequent ripening at ambient temperatures [29]. However, 30% or 60% CO_2_ treatment significantly reduced CI symptoms (based on pitting) in fruits stored for 14 d at 4 °C, followed by 8 d at 20 °C (Figure 2C). This indicated that CO_2_ treatment suppressed CI. In particular, 30% CO_2_ treatment prevented CI symptoms, such as surface pitting, more effectively than the higher order treatment.

The decay rate increased dramatically in control fruits transferred to ambient storage conditions at 20 °C after cold storage, whereas the group exposed to 60% CO_2_ had a significantly reduced decay rate (Figure 2D). Therefore, 30% or 60% CO_2_ treatments can be applied to reduce the decay rate during cold storage of tomatoes.

### 3.4. RNA-Seq and Functional Categorization of CO_2_-Responsive Genes

To understand how CO_2_ treatment regulates the physiological and biochemical modifications related to CI, a comparative transcriptomic analysis was performed in the pericarp tissue from CO_2_-treated (30% and 60%) and untreated tomatoes before cold storage and subsequent to subjecting the fruits to storage at ambient conditions. A large number of DEGs were identified from the 0 d CO_2_ treatment. The complete details of DEGs from all comparisons, including Venn diagrams, are provided in Appendix A. The heat map revealed dramatic changes that occurred after the CO_2_ treatments, which separated the control tomatoes at each storage time point (Appendix A).

To provide a functional overview of the DEGs between the CO_2_-treated and untreated groups, DAVID analysis was performed using all CO_2_-responsive genes. The Gene Ontology terms annotated for the DEGs belonged to 21 functional groups, including cellular components, biological processes, and molecular functions (Appendix A), implying that these DEGs are functionally involved in diverse physiological processes. In the KEGG database, the most abundant pathway (lowest *p*) was the biosynthesis of secondary metabolites (sly01110) (Figure 3A).

Conserved domains in the DEGs were further identified; in particular, the most abundant domain was the *ethylene response factor* (*ERF*) (Figure 3B). Therefore, we examined the effects of CO_2_ treatment on ethylene-related genes. Gene expression analysis showed that the ethylene signaling-related genes *ERF1*, *ERF2*, and *ERF4* were upregulated upon CO_2_ treatment at day 0. In contrast, the ethylene synthesis-related gene *1-aminocyclopropane-1-carboxylate synthase 4* (*ACS4*) was significantly downregulated during cold storage of CO_2_-treated fruits (Figure 4). However, CO_2_ treatment and cold storage, followed by ambient storage conditions at 20 °C, downregulated *ERF107*. These results suggest that CO_2_ treatment reduced ethylene synthesis and subsequently enhanced ethylene signaling to extend the shelf life of tomatoes.

### 3.5. Primary Metabolite Profiling for Pathway Analysis

Metabolites of untreated and CO_2_-treated tomatoes during postharvest storage for the treatment including 0 d and 14 d at 4 °C, and the treatment including 14 d at 4 °C followed by 8 d at 20 °C, were analyzed using GC-MS. The water-soluble metabolites were presented based on the KEGG pathway (Figure 5). To analyze the KEGG pathway using MetaboAnalyst, control and 30% CO_2_ treatment at the above time points were selected to characterize the treatment effect. No considerable difference was observed between 7 d and 14 d at 4 °C. Thus, metabolites from day 7 at 4 °C were not included here. The relative concentrations of sugar, valine, and glutamic acid significantly decreased after 30% CO_2_ treatment on day 0 (*p* < 0.01). Similarly, the relative concentrations of phenylalanine, tyrosine, aspartic acid, and lysine were significantly decreased after 30% CO_2_ treatment on day 0 (*p* < 0.05). The relative concentration of sucrose increased significantly in 30% CO_2_ treatment during 14 d storage at 4 °C (*p* < 0.05). Meanwhile, the relative concentrations of alanine, valine, lysine, glutamic acid, serine, glycine, and tyrosine were significantly increased in 30% CO_2_ treatment, followed by storage at 4 °C for 14 d and for 8 d at 20 °C (14 + 8 d) (*p* < 0.05). Relative concentrations of sucrose and glutamine were significantly increased in 30% CO_2_ treatment, followed by storage at 4 °C for 14 d and for 8 d at 20 °C (14 + 8 d) (*p* < 0.05), whereas the relative concentration of fructose was significantly decreased in 30% CO_2_ treatment, followed by storage at 4 °C for 14 d and for 8 d at 20 °C (14 + 8 d) (*p* < 0.05). Charts using non-normalized values are displayed in Appendix A.

PLS-DA was conducted using Pareto scaling from the GC-MS data. In the PLS-DA score plot (Figure 6A) of the 30% CO_2_ treatment, followed by storage at 4 °C for 14 d and for a period of 8 d at 20 °C (14 + 8 d), component 1 explained 51.7% of the total variance, and component 2 explained 15.9% of the total variance. The two treatments were tightly clustered and separated from each other on a score plot without overlap of the 95% confidence intervals. The PLS-DA loading plot (Figure 6B) of the 30% CO_2_ treatment, followed by storage at 4 °C for 14 d and for 8 d at 20 °C (14 + 8 d), showed that most metabolites were located on the left. Threonine, serine, tyrosine, glycine, beta-aminoisobutyric acid, valine, and leucine had high variable importance of projection (VIP) values (Figure 6C), indicating that these metabolites are useful biomarkers of tomato during postharvest storage for 14 d at 4 °C, followed by 8 d at 20 °C. Regarding the PLS-DA score plot (Appendix A) of the 60% CO_2_ treatment, followed by storage at 4 °C for 14 d and for 8 d at 20 °C (14 + 8 d), component 1 explained 54% of the total variance, and component 2 explained 15% of the total variance. Additionally, the PLS-DA loading plot (Appendix A) of the 60% CO_2_ treatment, followed by storage at 4 °C for 14 d and for 8 d at 20 °C (14 + 8 d), showed that most metabolites were located on the left. Tyrosine, leucine, aspartic acid, sucrose, and proline had high variable importance of projection values (Appendix A), indicating that these metabolites are biomarkers of tomato during postharvest storage for 14 d at 4 °C, followed by 8 d at 20 °C (14 + 8 d).

Pathway analysis was performed using metabolites from the 30% (Figure 6D) and 60% (Appendix A) CO_2_ treatments, followed by storage at 4 °C for 14 d and for 8 d at 20 °C (14 + 8 d). Three metabolic pathways (isoquinoline alkaloid biosynthesis; alanine, aspartate, and glutamate metabolism; and glycine, serine, and threonine metabolism) between the CO_2_-treated and untreated groups changed significantly and were characterized by −log_10_(*p*) value (>1.5) and impact value (0.3).

## 4. Discussion

### 4.1. Role of CO_2_ Treatment in the Response of Tomato Fruit to Low Temperature

In the present study, we found that short-term CO_2_ pretreatment delayed ripening and reduced CI symptoms, consequently extending the shelf life of tomatoes. The ripening process in tomato involves a complex and coordinated series of changes in pigmentation, flavor, texture, and aroma, resulting from physiological and biochemical activities. However, cold storage inhibits lycopene synthesis [21]. In the present study, CO_2_ pretreatment and cold storage synergistically inhibited lycopene development, resulting in a low a* value (Figure 2). Treatment with high concentrations of CO_2_ blocks or delays ripening by suppressing ripening-related gene expression [30]. Rugkong et al. [31] reported that uneven ripening in cold-stored tomatoes was related to the downregulation of genes associated with ethylene biosynthesis and signaling, which was reflected in the reduced ethylene production and lycopene accumulation observed in the experiment. In the present study, ethylene production was not significantly different between the CO_2_-treated and untreated tomatoes under cold-storage conditions (Figure 1). Nevertheless, it should be noted that evident changes in ethylene production due to CO_2_ treatment are difficult to observe because cold storage restricts ethylene production. Hence, CO_2_-induced delays in ripening have a synergistic relationship with cold storage.

Under ambient storage conditions at 20 °C, the CO_2_-treated tomatoes were significantly firmer and showed less ethylene production than the control fruits, implying the significant role of CO_2_ in fruit softening. CO_2_ treatment has been reported to increase firmness in peach by altering cell wall-degrading enzyme activity [12], and high CO_2_ levels were also found to influence cell-wall calcium binding, thereby increasing fruit firmness [32]. Furthermore, a recent report showed that CO_2_ treatment delayed cell wall degradation, thus maintaining the integrity of the middle lamella in strawberry and downregulating the level of the cell degradation enzyme, pectin esterase [10]. An increase in fruit firmness resulting from postharvest CO_2_ treatment occurs primarily through calcium-mediated pectin polymerization [33].

The CO_2_-treated tomatoes also showed a lower CI index than did the control (Figure 2C). CI is related to increased membrane permeability, increased leakage of ions from cells into intercellular spaces within tissues [2], and ultrastructural changes in the membrane [34]. Moreover, in the present study, functional analysis of DEGs showed that CO_2_-responsive genes were most significantly involved in the integral components of the plasma membrane, regulation of defense responses, and cell wall biogenesis (Appendix A).

### 4.2. CO_2_-Induced Global Transcriptional Changes

Comparative transcriptome analysis was performed to determine the mechanisms underlying CO_2_-altered fruit quality. The results showed dramatic transcriptomic changes between CO_2_-treated and untreated tomatoes. Interestingly, functional analysis showed that the major domain of the DEGs was ERF (Figure 3B). ERFs are known to participate in the last step of the ethylene signal transduction pathway and play important roles in the fruit ripening process and abiotic stress response [35,36]. ERF proteins play an important role in cold response by regulating the expression of downstream stress-related genes [37]. In tomato, 77 ERFs have been identified, 19 of which are related to ripening [38]. Most tomato ERFs, such as *LeERF*, *Pti*, and *JERF*, are responsive to environmental stresses, including low temperature, wounding, and salinity [32,39,40]. The ERFs identified in the present study responded differently at each time point. While *ERF1*, *ERF2*, and *ERF4* were upregulated by CO_2_ treatment at day 0, *ERF107* was downregulated under storage conditions at 20 °C after cold storage (14 + 8 d) (Figure 4). Romero et al. [41] reported that *ERF* genes play a role in the beneficial effect of high CO_2_ levels on the maintenance of table grape quality during storage at low temperatures, whereas *VviERF2c* may play a role in modulating *PR* gene expression. ERFs involved in CI are reduced by methyl jasmonate, suggesting that *ERF1* plays a role in regulating CI [42]. The present study suggests that CO_2_ treatment triggered ethylene signaling, especially involving ERFs that regulate cold stress, and reduced CI in tomato.

1-Aminocyclopropane-1-carboxylate synthase (ACS) is a key enzyme in ethylene biosynthesis and in the regulation of the transition from system-1 to system-2 ethylene synthesis in tomato. Previous RNA-Seq analysis using the ‘Micro Tom’ variety of tomato showed that chilling blocked the second step of ethylene biosynthesis [1]. Another study revealed that the regulation of *TERF2*/*LeERF2* is associated with enhanced freezing tolerance in tobacco and tomato through ethylene biosynthesis [37].

*LeACS2* and *LeACS4* expression levels increase during tomato fruit ripening [43]. In the present study, the ethylene synthesis gene, *ACS4*, was significantly downregulated by CO_2_ treatment (Figure 4). This suggests that delayed ripening occurs via the blocking of ethylene synthesis by both CO_2_ treatment and cold storage. Our findings suggest that CO_2_ treatment reduced CI symptoms and delayed ripening by regulating both ethylene synthesis and ethylene signaling, especially involving ERFs, which, in turn, control other downstream factors. However, further research is required to elucidate the link between ERFs and the downstream factors that control CI occurrence.

### 4.3. Effect of CO_2_ Treatment on Tomato Metabolites and Quality

The tomatoes that underwent short-term CO_2_ treatment showed significant changes in the metabolites involved in starch and sucrose metabolism (*p* = 0.03, impact = 0.39). In the present study, the sucrose concentration of tomatoes treated with CO_2_ over a short period was significantly higher than that of untreated tomatoes. A previous study reported that acid invertase activity significantly affects the ratio of sucrose to monosaccharides, including glucose and fructose [44]. In our study, the soluble acid invertase activity of short-term CO_2_-treated tomatoes might have decreased, along with a reduction in the respiration rate (glycolysis). This result is also consistent with the highly enriched glycolysis/gluconeogenesis from KEGG pathway enrichment analysis based on transcriptome analysis (Figure 3). We also found that the levels of malic acid and citric acid in short-term CO_2_-treated tomatoes were not significantly different from those in the untreated ones. Sangwanangkul et al. [13] reported several organic acid concentration changes in cherry tomatoes after treatment with 20% and 60% CO_2_. Although a significant decay rate was observed in the 60% CO_2_ treatment at 12 °C, no specific organic acid trend was observed after the CO_2_ treatment. According to an experiment by Centeno et al. [45], malate metabolism in tomatoes significantly affected sugar and starch metabolism. Consequently, the modified malate metabolism changed resistance to *Botrytis cinerea* through increased wrinkling. However, we did not observe any significant changes in the malate metabolism. Taken together, the improved quality of tomato fruits may be due to the synergistic effect of CO_2_ pretreatment before cold storage.

Four metabolic pathways were significantly affected by CO_2_ treatment. Isoquinoline alkaloid biosynthesis and alanine, aspartate, and glutamate metabolism have been reported as drought stress-related metabolic pathways in drought stress experiments [46,47]. Water loss during postharvest storage may be associated with drought stress. Glutamate is the first amino acid related to nitrogen fixation and is decreased by drought stress because plants cannot take up nitrogen fertilizer from the soil under this condition. During postharvest storage, fruits cannot obtain exogenous nitrogen while maintaining cellular activity, which includes consuming energy and converting compounds. Thus, control tomatoes with higher levels of respiration could have spent amino acids rapidly, resulting in differences in alanine, aspartate, and glutamate metabolism. After 14 d of cold storage and 8 d of storage at ambient conditions at 20 °C, 10 different amino acids were observed to be at higher concentrations in CO_2_-treated tomatoes than in the control tomatoes. These included serine, glycine, glutamate, glutamine, tyrosine, alanine, leucine, lysine, threonine, and isoleucine. Treatment with CO_2_ significantly changed amino acid and sugar concentrations after cold storage and storage at ambient conditions, indicating that CO_2_-treated tomatoes might have a better taste than untreated tomatoes.

Biosynthesis of secondary metabolism was highly enriched in KEGG pathway enrichment analysis based on the transcriptome (Figure 3A). This result may be related to carotenoid biosynthesis, as 60% CO_2_ treatment significantly delayed lycopene biosynthesis at 14 d and 14 d plus 8 d storage at ambient conditions at 20 °C. Plant–pathogen interactions are mostly related to secondary metabolism. Flavonoid biosynthesis starts from the shikimate pathway, with phenylalanine as a substrate. Thus, the selected DEGs explained complementarity to each other. A previous study [48] reported that quercetin and rutin were the most abundant flavonoids in tomatoes. These flavonoids gradually increased from the breaker stage to the pink or light red stages, depending on the cultivar.

## 5. Conclusions

In summary, the present study showed that CO_2_ treatment before cold storage reduced CI symptoms and extended shelf life by delaying ripening in tomato. This technique can be applied to tomatoes that require long-term cold chain transportation. In addition, transcriptome and metabolome profiling provided basic physiochemical information underlying the response of tomato to CO_2_ treatment. Transcriptome and metabolome profiling indicated altered ethylene biosynthesis and ethylene signaling via ERFs, and respiration-related pathways appeared to control CO_2_-induced fruit modifications. This suggests that CO_2_ treatment delayed fruit ripening by regulating carbohydrate metabolism and ethylene-related genes. Therefore, our findings will help in developing strategies to reduce CI symptoms and extend the shelf life of other subtropical crops.

## Figures and Tables

**Figure 1 foods-10-00744-f001:**
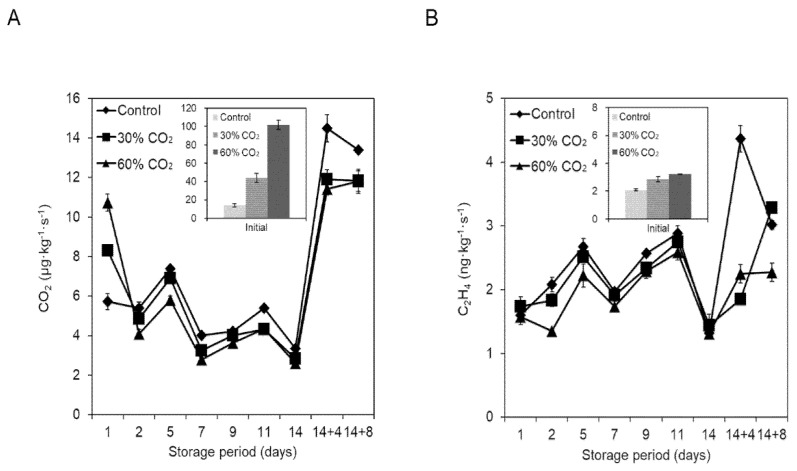
(**A**) Respiration rate and (**B**) ethylene production in CO_2_-treated tomatoes during storage. Samples were obtained from untreated (control) and CO_2_-treated tomatoes during storage at 4 °C for 14 d and were transferred to 20 °C for another 4 (14 + 4) to 8 d (14 + 8). The graph denoted ‘Initial’ (inner box) indicates control or CO_2_-treated tomatoes before cold storage. Data are shown as the mean ± standard error of three replicates.

**Figure 2 foods-10-00744-f002:**
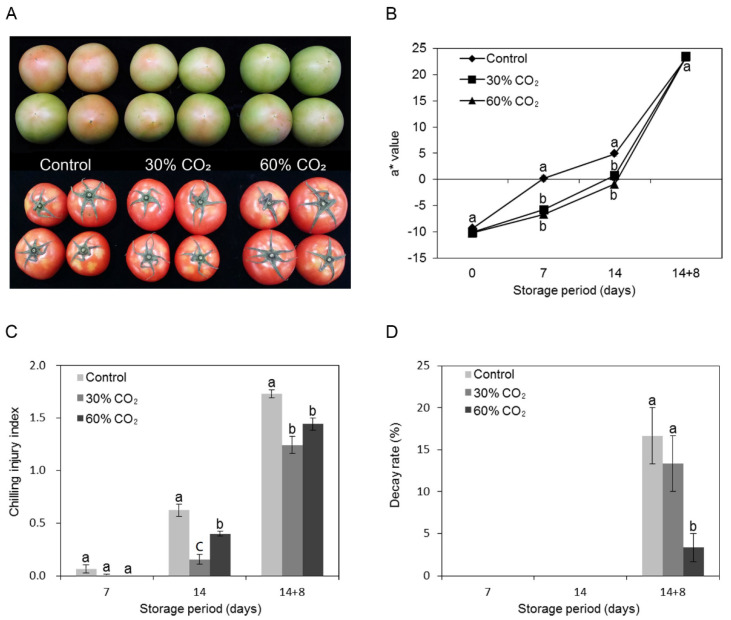
Effect of short-term CO_2_ treatment on tomato quality. (**A**) Photographs of CO_2_-treated and untreated tomatoes taken after storage for 7 d (upper) and 14 d at 4 °C followed by storage at 20 °C ambient conditions for a further 8 d (14 + 8; lower). (**B**) Changes in the skin color a* value (redness) of CO_2_-treated or untreated tomatoes during the cold storage and after storage at ambient conditions. Values are the means of 15 replicate samples ± standard error (SE). (**C**) Chilling injury index of tomato. (**D**) Decay rate of tomato. Data are shown as the mean ± SE of three replicates. Untreated tomatoes were used as the control. Different letters indicate significant difference among treatments within the same storage period by Duncan’s multiple range test. with *p* < 0.05.

**Figure 3 foods-10-00744-f003:**
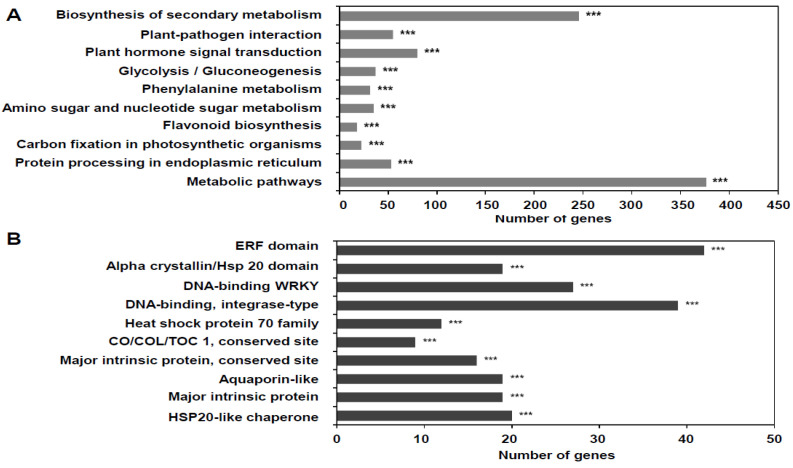
Global analysis of differentially expressed genes (DEGs) in CO_2_-treated tomatoes. (**A**) KEGG pathway enrichment analysis of the DEGs. (**B**) InterPro domain analysis of DEGs. Samples were obtained from untreated (control) and CO_2_-treated tomatoes. Analyses were performed using DAVID v.6.8. *** represents *p* ≤ 0.001.

**Figure 4 foods-10-00744-f004:**
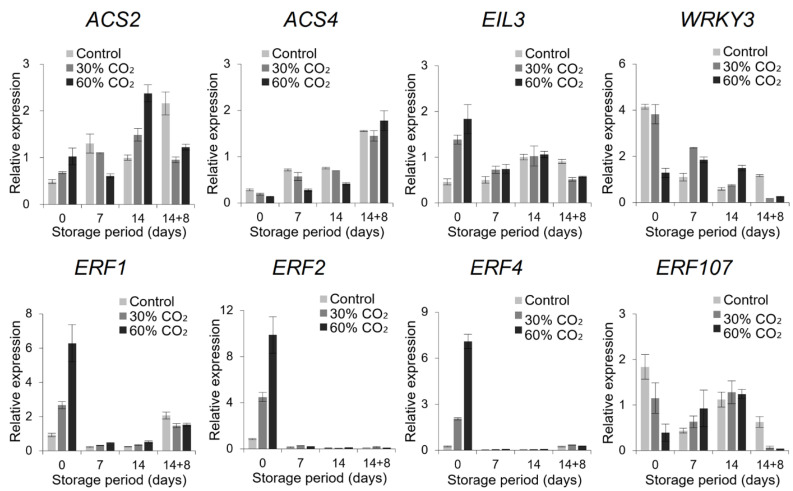
Quantitative real-time-PCR validation of differentially expressed genes (DEGs) identified by RNA-Seq analysis. The expression of selected DEGs, including the ethylene-related genes and *WRKY*, was examined using qRT-PCR. Samples were obtained from tomatoes treated or untreated with CO_2_ after 0 d, 7 d, and 14 d at 4 °C and after 14 d at 4 °C followed by 8 d at 20 °C (14 + 8 d). The bar represents the mean ± standard error of three biological replicates.

**Figure 5 foods-10-00744-f005:**
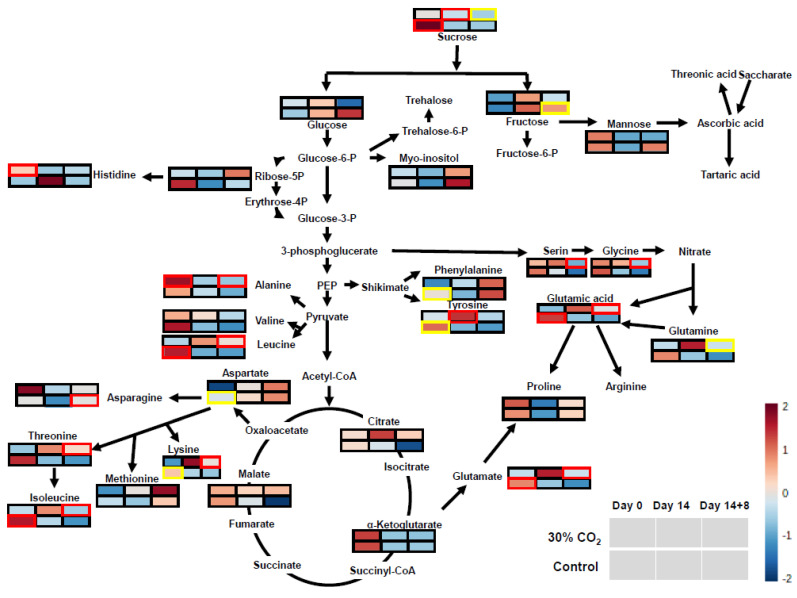
Map of metabolites involved in the sugar, amino acid, and tricarboxylic acid (TCA) metabolism pathways. Map of metabolites in red and blue represent those upregulated and downregulated, respectively, with no treatment and CO_2_ treatment. Red (*p* = 0.01) and yellow (*p* = 0.05) boxes represent significantly different *p*-values by Tukey’s HSD. The relative levels of expression in the control and 30% CO_2_ treatment groups at 0, 14, and 14 + 8 d are shown as a heat map.

**Figure 6 foods-10-00744-f006:**
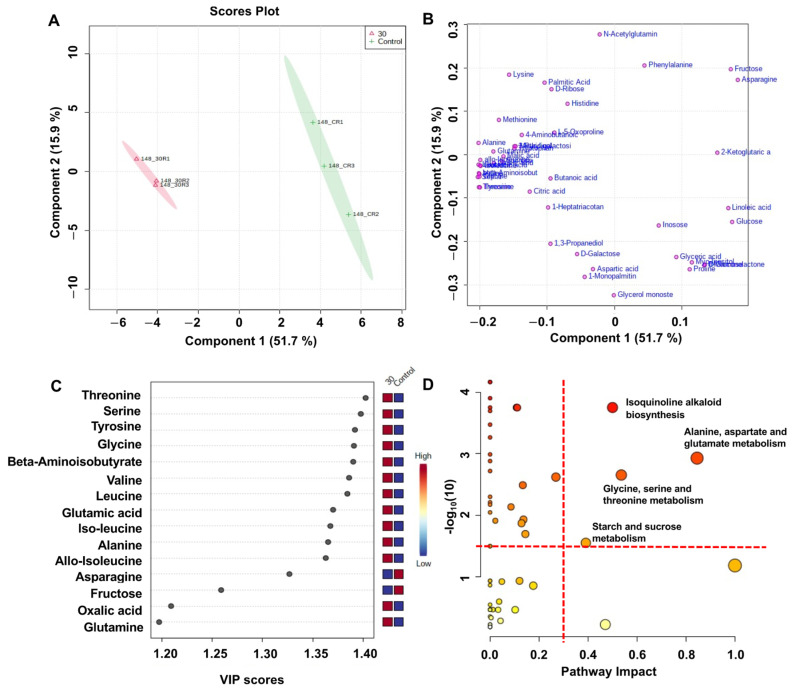
Partial least squares-discriminant analysis (PLS-DA) (**A**) score plot and (**B**) loading plot, derived from gas chromatography-mass spectrometry (GC-MS) data of untreated and 30% CO_2_-treated tomatoes during postharvest storage for 14 d at 4 °C, followed by 8 d at 20 °C (14 + 8 d). (**C**) PLS-DA variable importance of projection (VIP) score derived from GC-MS data of untreated and 30% CO_2_-treated tomatoes during postharvest storage for 14 d at 4 °C, followed by 8 d at 20 °C (14 + 8 d). (**D**) Top 15 VIP scores of metabolites characterized by VIP scores (>1.238) in the KEGG pathway analysis from untreated and 30% CO_2_-treated tomatoes during postharvest storage for 14 d at 4 °C, followed by 8 d at 20 °C (14 + 8 d). The named pathways in bold represent the significantly changed metabolism pathways characterized by −log_10_(*p*) value (>1.5) and impact value (0.3).

**Table 1 foods-10-00744-t001:** Firmness, solid soluble content (SSC), titratable acidity (TA), and pH of tomatoes treated or untreated with CO_2_ and stored for 0 d, 7 d, or 14 d at 4 °C or stored at 4 °C for 14 d followed by 8 d at 20 °C (14 + 8 d).

Treatment	0 d	7 d	14 d	14 + 8 d
	*Firmness (N)*			
Control	19.83 ± 0.41 Aa ^1^	16.56 ± 0.18 Ab	13.67 ± 0.34 Ac	6.54 ± 0.38 Bd
30% CO_2_	19.94 ± 0.47 Aa	15.88 ± 1.23 Ab	13.53 ± 0.28 Ac	8.07 ± 0.50 Ad
60% CO_2_	20.42 ± 0.67 Aa	17.84 ± 0.70 Ab	13.19 ± 0.57 Ac	7.94 ± 0.51 Ad
	***SSC (%)***			
Control	4.28 ± 0.08 Aa	4.46 ± 0.04 Aa	4.48 ± 0.02 Aa	4.48 ± 0.07 Aa
30% CO_2_	4.42 ± 0.10 Aa	4.44 ± 0.07 Aa	4.46 ± 0.07 Aa	4.50 ± 0.06 Aa
60% CO_2_	4.26 ± 0.07 Ab	4.50 ± 0.03 Aa	4.38 ± 0.04 Aab	4.40 ± 0.04 Aa
	***TA (%)***			
Control	1.07 ± 0.01 Aa	1.05 ± 0.02 Aa	0.79 ± 0.05 Ab	0.66 ± 0.01 Ac
30% CO_2_	0.94 ± 0.02 Aa	0.99 ± 0.02 ABa	0.85 ± 0.03 Ab	0.68 ± 0.01 Ac
60% CO_2_	1.08 ± 0.01 Aa	0.98 ± 0.02 Ab	0.83 ± 0.02 Ac	0.65 ± 0.02 Ad
	***pH***			
Control	3.99 ± 0.02 Ab	3.97 ± 0.01 Bb	3.95 ± 0.03 Ab	4.32 ± 0.04 Aa
30% CO_2_	3.99 ± 0.02 Ac	4.01 ± 0.02 ABbc	4.07 ± 0.04 Ab	4.34 ± 0.02 Aa
60% CO_2_	3.93 ± 0.02 Ac	4.05 ± 0.02 Ab	4.05 ± 0.04 Ab	4.40 ± 0.02 Aa

^1^ Values represent the mean of 15 replicate samples ± standard errors. The same uppercase letter within each column, or the same lowercase letter within each row, indicates means that are not significantly different at *p* < 0.05, according to Duncan’s multiple range test.

**Table 2 foods-10-00744-t002:** Changes in the carotenoid content of tomatoes treated or untreated with CO_2_ and stored for 0 d, 7 d, or 14 d at 4 °C or stored for 14 d at 4 °C followed by 8 d at 20 °C.

Treatment	0 d	7 d	14 d	14 + 8 d
	*Lutein* (mg kg^−1^)			
Control	27.84 ± 4.16 Aa ^1^	24.45 ± 2.95 Ba	10.94 ± 1.32 Ab	8.85 ± 0.36 Ab
30% CO_2_	32.45 ± 1.95 Aa	34.94 ± 4.55 ABa	11.88 ± 2.06 Ab	8.55 ± 0.32 Ab
60% CO_2_	33.80 ± 1.85 Aa	36.25 ± 0.01 Aa	11.60 ± 0.63 Ab	8.61 ± 0.64 Ab
	***Lycopene* (mg kg^−1^)**			
Control	5.05 ± 0.43 Ab	4.82 ± 0.69 Ab	10.64 ± 1.79 Ab	39.56 ± 3.87 Aa
30% CO_2_	6.64 ± 1.54 Ab	4.74 ± 0.72 Ab	6.45 ± 1.48 ABb	26.88 ± 0.73 Ba
60% CO_2_	6.21 ± 0.12 Ab	3.37 ± 0.44 Ab	4.72 ± 0.18 Bb	26.57 ± 2.21 Ba
	***Beta-Carotene* (mg kg^−1^)**			
Control	15.06 ± 2.84 Aa	9.64 ± 1.88 Bab	7.38 ± 0.80 Ab	10.24 ± 0.18 Aab
30% CO_2_	14.67 ± 0.79 Aab	15.92 ± 2.28 Aa	9.73 ± 2.54 Abc	8.36 ± 0.29 Ac
60% CO_2_	17.02 ± 2.30 Aa	16.26 ± 0.64 Aa	7.51 ± 0.97 Ab	9.33 ± 0.91 Ab

^1^ Values represent the mean ± standard error of 15 replicate samples. The same uppercase letter within each column, or the same lowercase letter within each row, indicates means that are not significantly different at *p* < 0.05, according to Duncan’s multiple range test.

## Data Availability

Not applicable.

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
