# Peer review of "Carbon Dioxide Pretreatment and Cold Storage Synergistically Delay Tomato Ripening through Transcriptional Change in Ethylene-Related Genes and Respiration-Related Metabolism"

_foods, 2021, doi:10.3390/foods10040744_

Round 1

Reviewer 1 Report

there are quite some typing and grammatical errors in the MS, pls check

73  plants = fruit

101. total SSC? is total sol solids meant? or sol solids content

the differences in fig 1 are all very minor. although from stat point of view there may be differences, from a physiological point of view they seem irrelevant

192, do not think we are looking at a climacteric peak. the temp changes and that may be the reason that also ethylene prod changes as it will be higher at higher temp

the main effects of CO2 treatments seem a slower ripening (color. lycopene, firmness)  and less decay. Not clear how important the effect is? mostly te metabolic changes due to 30% CO2 are discussed, but this treatment has minor effects on CI index and on decay. Does the CO2 really make a difference?

about the metabolites, there are here and there sig differences due to the CO2 treatment, but I would like to also get indication about absolute levels (all data are normalised). are these differences also sign from a biological point of view? The changes in carbohydrate metabolism, how big are the differences between treated and non treated?

348  text says that sucrose was sign different but figure does not show a yellow or red box? In general only few metabolites show stat diff between treatmenst. I wonder how big differences are. so pls show e.g. in supplement

the discussion is quite long and partly repetition of results. can it be shortened to stress only the main findings

Author Response

Thank you for your comments.

We did our best to revise based on your comments.

Reviewer 2 Report

Dear authors, I suggest you to revise the introduction of your paper, as I do not consider it very original and innovative, for the reason that there are already several works that consider the effect of CO2 on tomato shelf life. With this regard, assuming that the publisher considers to publish your paper, I suggest you to strongly modify the introduction and bibliographic references, taking into account the numerous scientific evidence of international stature, on the subject.
Some examples can be easily found in common scientific databases.
In addition, regarding the content, I propose the following revisions:
line 87
Please, authors clarify the following sentence, where it's not clear and/or. Some samples were directly stored at room temperature without the cold storage?

The fruits were stored in a commercial cardboard box covered with plastic film at
87 4°C (cold storage) for 14 d and/or transferred to 20°C (shelf-life conditions) for 8 d

Lines 90-95
Please, authors add a reference for the method used in the gas analysis of Ethylene

Line 98
Authors could change "colour differences meter" in "colour meter".

Line 99
Please, authors change "and reported based on Hunter's scale" with "and values were reported based on Hunter's scale".

Line 118
Please, authors add a reference for the method used in the HPLC analysis method.

Line 132
add a space between v1.3.3b and [19].

Author Response

(The authors gave the same response as above.)

Reviewer 3 Report

The article "Carbon dioxide pretreatment on tomatoes before cold storage 2 synergistically delays ripening through transcriptional change 3 of ethylene-related genes and respiration-related metabolisms" is very interesting and has been studied from several chemical and physical aspects.

The aspect of research are different and were also very laborious. However, I think that the research is referred to two treatments while the metabolomic analysis is referred only to one treatment.
How come?Can you explain the motivation of this choice? Could you explain it better? 

 In order to improve the article and the reading I suggest you to create a table with the initial chemical, physical and biochemical values of the tomato fruit at harvest including the color L*a*b

Among the treatments used there are control, 30% CO2-treated, and 60% CO2-treated. 
Explain why you chose to show only one treatment in  results and  discussion.
I suggest you expand and discuss the conclusions better. You should also be more incisive in the final part of the abstract.

Author Response

(The authors gave the same response as above.)

Round 2

Reviewer 2 Report

Dear Authors,

no other changes are required.

Author Response

Thank you.